# Synthetic Data Generated from CT Scans for Patient Pose Assessment

**Manuel Laufer**[1]                                                             M.LAUFER@UNI-LUEBECK.DE
**Dominik Mairhöfer**[1]                                             D.MAIRHOEFER@UNI-LUEBECK.DE
**Malte Sieren**[2]                                                                 MALTE.SIEREN@UKSH.DE
**Hauke Gerdes**[2]                                                               HAUKE.GERDES@UKSH.DE
**Fabio Leal dos Reis**[2]                                                 FABIO.LEALDOSREIS@UKSH.DE
**Arpad Bischof**[2,3]                                                           ARPAD.BISCHOF@UKSH.DE
**Thomas Käster**[4]                                                                     TK@PRCMAIL.DE
**Erhardt Barth**[1]                                                     ERHARDT.BARTH@UNI-LUEBECK.DE
**Jörg Barkhausen**[2]                                                   JOERG.BARKHAUSEN@UKSH.DE
**Thomas Martinetz**[1]                                         THOMAS.MARTINETZ@UNI-LUEBECK.DE

[1] *Institute for Neuro- and Bioinformatics, University of Lübeck, Germany*

[2] *University Medical Center Schleswig-Holstein, Lübeck, Germany*

[3] *IMAGE Information Systems Europe GmbH, Rostock, Germany*

[4] *Pattern Recognition Company GmbH, Lübeck, Germany*

**Editors:** Accepted for publication at MIDL 2025

## Abstract

An adequate diagnostic quality of radiographs is essential for reliable diagnoses and treatment planning. The patient's pose during radiography is one of the most important factors determining the diagnostic quality. Since patient positioning is difficult and not standardized, an automated AI-based approach using depth images to automatically assess the patient's pose before the radiograph has been taken would be helpful. Due to regulatory hurdles, however, it is difficult in practice to acquire the required depth images and corresponding radiographs. In this paper, we present a framework that can generate such training data synthetically from Computer Tomography scans. We further show that by pretraining on our generated synthetic dataset consisting of 3077 image pairs of upper ankle joints, the pose assessment of real upper ankle joints can be improved by up to 11 percentage points.

**Keywords:** patient pose assessment, synthetic data generation, diagnostic quality, CT scan, time-of-flight cameras, radiography, deep learning

## 1. Introduction

The diagnostic quality of radiographs is essential for making reliable diagnoses and planning treatments. Radiographs of inadequate diagnostic quality often lead to retakes and thus to increased radiation exposure for the patient and increased costs for the hospital. In the worst case, inadequate diagnostic quality can lead to incorrect treatment and misdiagnosis. The most important factor affecting the diagnostic quality of a radiograph is the pose of the patient at the time the radiograph is taken (Little et al., 2017). Furthermore, patient positioning is error-prone, as it is not standardized and depends heavily on the patient and the experience of the radiographer, who is also often under time pressure.

To assist the radiographer in positioning the patient and to protect the patient from increased radiation dose, an automatic pose assessment would help. By attaching two Time-of-Flight (ToF) cameras to the X-ray device, we were able to show recently that depth images of anatomical preparations of upper ankle joints contain information that can lead to high accuracy pose assessment (Laufer et al., 2024). In order to determine a correspondence between the depth image of the pose and the diagnostic quality of the radiograph, the radiograph and the depth image must be taken simultaneously and labeled with their diagnostic quality. The depth image and the label can then be used to train neural networks to predict the diagnostic quality of the radiograph before the radiograph is even taken. However, radiographing subjects without an indication is problematic. In particular, intentionally radiographing subjects in non-diagnostic poses, which are necessary for the training, is ethically difficult to justify. Furthermore, using cameras in live clinical practice is not readily possible for data protection and regulatory reasons. Finally, working with anatomical preparations as a solution is not scalable.

To address this challenge, we present a framework that synthetically generates the required image pairs of depth images and radiographs from Computer Tomography (CT) scans. CT scans that have already been taken can thus be used retrospectively to create a dataset of any size, which makes the approach scalable. It is furthermore possible to intentionally generate non-diagnostic poses by selectively adjusting the CT scans. We show that by pretraining on our generated synthetic dataset of upper ankle joints, the pose assessment of real upper ankle joints can be improved by up to 11 percentage points (pp). The dataset is published under https://github.com/INB-KI-SIGS/patient-pose-assessment.

## 2. Related Work

For the generation of synthetic radiographs from CT scans, known as digitally reconstructed radiographs (DRR), two main approaches are used: The ray tracing approach, using forward projection, casts a ray through the CT volume for each pixel on the detector and accumulates the intensity of the values along the path. Although ray tracing is computationally efficient, it is not capable of modeling scattering or beam hardening (Russakoff et al., 2005). In the approach presented in Unberath et al. (2018), a radiograph generated by forward projection is combined with deep learning-based scatter and noise estimation. In contrast, the Monte Carlo (MC) approach simulates the transport of photons across the CT scan to model the photon-matter interaction and therefore requires material properties for each CT voxel (Badal and Badano, 2009). Such simulations result in realistic DDRs; however, they are computationally more expensive than forward projection. Although there is little research on generating synthetic depth images from CT scans, more works are investigating the generation of point clouds from CT scans. Chougule et al. (2013) present a slice-based approach using automatic thresholding and edge detection to generate point clouds from CT scans. A voxel-based approach to generate surfaces from medical 3D data is the Marching Cubes Algorithm (MCA) (Lorensen and Cline, 1987). Saiti and Theoharis (2022) use the MCA to create synthetic point clouds from CT scans to learn multimodal registration with point clouds and CT scans. To the best of our knowledge, there is no framework that generates pairs of synthetic depth images *and* corresponding radiographs for different views from CT scans and uses them for training patient pose assessment models.

## 3. Framework

The generation of synthetic radiographs and depth images from CT scans involves several steps. First, the surface of the target anatomy is extracted from the CT scan as a point cloud. The point cloud is then augmented to simulate different body types. These point clouds are placed on a pre-recorded point cloud of an imaging table in an X-ray room and rotated to create poses of different diagnostic quality. A synthetic depth image is generated for each pose by 2D projections of the point clouds. The corresponding synthetic radiograph is generated for each pose from the CT scan using a MC simulation. Our framework, which is implemented via Open3d's (Zhou et al., 2018) graphical visualization, is shown in Figure 1 and examples of synthetically generated depth images and radiographs are shown in Figure 4 in Appendix C.

### 3.1. Preprocessing

In order to generate a synthetic depth image of the target anatomy from a CT scan, the target anatomy must first be extracted from the CT. For this, the scan is converted to a point cloud using MCA. The threshold value of the MCA is set to -500 Hounsfield units (HU) so that the air around the patient is removed. This conversion from a CT array to a point cloud includes a transformation $T_{CT}^P$ from the CT coordinate system $CT$ to the patient coordinate system $P$. The point cloud is then cropped to the target anatomy to simplify the subsequent steps and calculations. Since only the surface of the target anatomy is relevant for the synthetic depth image, additional MCA runs, the clustering algorithm DBSCAN (Ester et al., 1996), and cropping are applied to remove the imaging table and points that do not belong to the surface of the target anatomy. In order to generate poses with different diagnostic quality, including inadequate quality, the target anatomy can be brought into other poses by specific rotations of the point cloud. The axis of rotation is strongly dependent on the target anatomy. For the upper ankle joint, it is the longitudinal axis, which is positioned in the point cloud such as to mimic human leg rotations. This axis passes through the center of the upper ankle joint. See Figure 3(a) in Appendix B for illustrations.

### 3.2. Augmentation

By determining the normal vectors of the point cloud, it is possible to shift the point cloud both outwards and inwards along the direction of the normal vector, resulting in two additional point clouds. This allows us to simulate patients with different shapes and to increase the amount of data; see Figure 3(b) in Appendix B. The diagnostic-quality label applies to all augmentations of a particular pose, as it can be assumed that the anatomy that determines the quality, in particular the position of the bones in relation to each other, does not change with minor displacements along the normal direction.

### 3.3. Scene Composition and Synthetic Depth Image Generation

To generate realistic synthetic depth images, it is beneficial to embed the target anatomy in a realistic scene. This can be achieved by recording the X-ray room in advance, including the imaging table, so that the target anatomy can then be placed on the imaging table

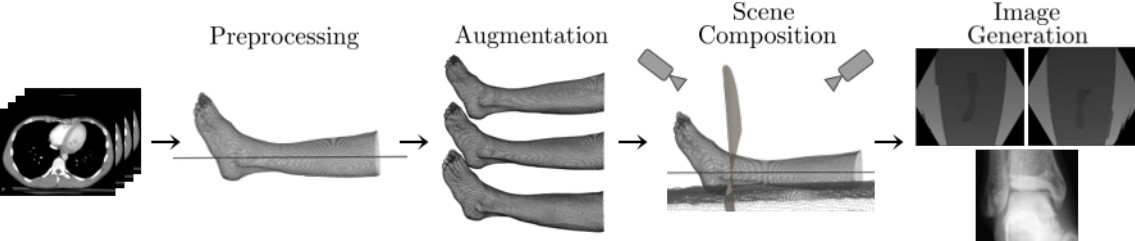

Figure 1: Schematic overview of the framework. The CT scan as input is passed through the steps described in Section 3 to generate synthetic radiographs and depth images.

under the X-ray device; see Figure $3(c)$ in Appendix B. The exact position of the target anatomy is selected so that the X-ray beam of the X-ray device passes through the axis of rotation of the target anatomy. Since a realistic environment and positioning of the target anatomy has been established, it is possible for the user to easily determine the range of rotation of the target anatomy to include non-diagnostic poses. Both the rotation and the initial embedding of the target anatomy in the X-ray room can be described as a further transformation $T_P^{ToF}$ from $P$ to the ToF coordinate system $ToF$. From the point clouds of the target anatomy and the X-ray room, a 2D projection yields the synthetic depth image by taking into account intrinsic and extrinsic camera parameters and distortion coefficients. This way, synthetic depth images can be generated for any desired angles of rotation, augmentations, and camera views.

### 3.4. Synthetic Radiograph Generation

When generating synthetic radiographs, it is important that all depicted anatomical features are identical to those of real radiographs in the same position to avoid false labels of the corresponding depth images. Generation based on a physical model was therefore preferred to methods based on deep learning. Since only a region smaller than the one depicted in the point clouds described in Section 3.1 must be visible in the radiograph, the target anatomy is cropped out of the CT in the first step. The cropped CT voxels are then converted into material and mass-density voxels. The material voxels each contain one of the materials air, soft tissue, bone, or titanium, based on the HU value of the respective voxel. Similarly, the mass-density voxels contain the density of the material adjusted by the HU value.

For each pair of material and mass-density voxels, multiple radiographs corresponding to different positions are generated. Instead of changing the position of the target anatomy, which would require rotation and interpolation of the voxels, the position of the X-ray device relative to the target anatomy is changed; see Figure $3(d)$ in Appendix B. The corresponding position of the X-ray device in the CT coordinate system can be obtained using the inverted transformations $(T_P^{ToF})^{-1}(T_{CT}^P)^{-1}$. To generate the radiograph, the MCGPU tool (Badal and Badano., 2011) was used for a MC simulation together with the material properties from the PENELOPE 2006 material files (Salvat et al., 2006) to simulate $2 \cdot 10^{10}$ X-ray beam paths. While more simulated paths would reduce the noise in the generated radiographs, the simulation time would increase. The use of $2 \cdot 10^{10}$ simulated paths allowed us to

create realistic radiographs in a reasonable amount of time. The resulting raw image is then converted to a synthetic radiograph using a non-linear value mapping, to obtain a look similar to a real radiograph. The synthetic radiograph is not as detailed as a real radiograph, but expert radiologists have validated that the visual quality is suitable for assessing the diagnostic quality for a given pose.

## 4. Datasets

The two datasets used in this paper consist of depth images from two camera views and corresponding radiographs. Each of these radiographs was assessed by 4 radiologists to determine its diagnostic quality on a scale of 1 to 3 in steps of 0.5. A diagnostic quality of 1 is ideal and a diagnostic quality of 3 is inadequate. The deciding factor in the assessment of the upper ankle joint is the visibility of the joint space; see Figure 4 in Appendix C. Radiographs with a label in the interval of [1,2.5) can be further classified as *diagnostic* and anything with a label greater than that as *non-diagnostic*.

### 4.1. Synthetic Dataset

Using the framework proposed in Section 3, we were able to generate pairs of synthetic radiographs and depth images of upper ankle joints from 10 CTs of different patients in 3077 different poses. The anonymized CT scans were selected to contain flexed upper ankle positions and exclude clutter such as tubes or screws. From the 10 CTs, a total of 17 upper ankle joints were extracted and rotated medially around the longitudinal axis in a range of 90 degrees. A synthetic depth image was generated from two camera views $V_1$ and $V_2$ for each half degree, i.e. a total of 181 poses per foot. This was done for each of the three augmentations, resulting in a total of 18462 depth images. Since the synthetic radiograph is the same for all camera views and augmentations, one synthetic radiograph was created for each of the 181 poses resulting in a total of 3077 synthetic radiographs. To the best of our knowledge, this is by far the largest dataset linking depth images to diagnostic quality.

### 4.2. Anatomical Preparations Dataset

As presented in Laufer et al. (2024), we captured two anatomical preparations – a left and a right lower leg of two women in 174 different poses using two ToF cameras. Parallel to the depth images from two different views, a radiograph of the upper ankle joint was also taken. The preparations were not only rotated medially around the longitudinal axis but also flexed in three different positions of the ankle joint. More detailed information on this published dataset can be found in Laufer et al. (2024).

## 5. Experiments and Results

The two experiments carried out are designed to answer the following questions:

**Experiment 1**: Can a neural network be trained on the generated synthetic depth images to assess the pose with high accuracy?

**Experiment 2**: If so, can the neural network be finetuned on real data in order to improve the results of patient pose assessment?

Both experiments were modeled as regressions, to better reflect the nature of the labels. While the depth images served as input for the models, the models output a single continuous value between 1 and 3 as the diagnostic quality. All experiments were implemented using PyTorch and repeated 10 times with different seeds. The results were then averaged.

### 5.1. Experiment 1

The first experiment tests whether relevant features can be learned with only the synthetic dataset. In addition, to clarify whether the augmentation of the depth images (see Section 3.2) has a benefit, the training was carried out with and without augmentation.

**Training**    Two EfficientNet-B0 (Tan and Le, 2019), called $CNN_{V_1}$ and $CNN_{V_2}$ were used. Following the separate network architecture (see Figure 2(a)) described in Laufer et al. (2024) the $CNN_{V_1}$ is only trained on the depth images from camera view $V_1$ while the $CNN_{V_2}$ is trained on images from camera view $V_2$. The models each receive a single channel depth image $i$, resized to $336 \times 336$ pixels, as input and output a single continuous value $y$. The models are trained to minimize the Mean Squared Error (MSE; see Equation (1) in Appendix A) between the model's output $y$ and the averaged diagnostic quality $x$ that had been assigned to the input depth image. The Adam optimizer (Kingma and Ba, 2017) was used with an initial learning rate of $10^{-3}$, which was decreased 4 times by a factor of 10 in the last 10,000 steps. We used a batch size of 16 and trained each model for 200,000 steps. For this experiment, a three-fold cross-validation was performed on the synthetic dataset. For each cross-validation run the test set consisted of three randomly selected upper ankle joints so that no subject from the test set would appear in the training set. Note that possible differences in the results in comparison to Laufer et al. (2024) are due to different implementations and updated libraries. The results are shown in Table 1.

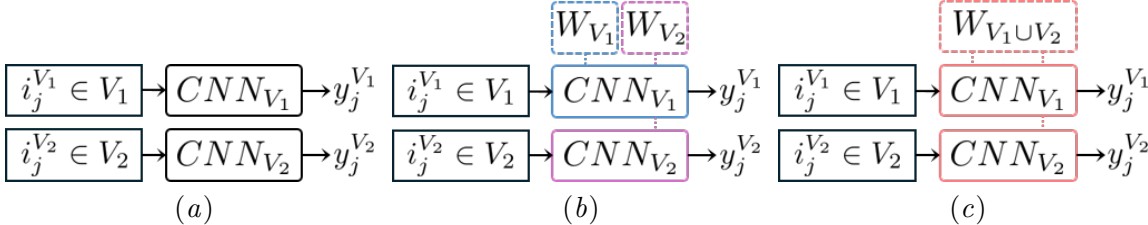

Figure 2: 2(a) shows the separate network architecture: individual images $i_j^{V_1}$ and $i_j^{V_2}$ from each camera view $V_1$ and $V_2$ are used to train two separate CNNs. $y_j^{V_1}$ and $y_j^{V_2}$ are the continuous outputs of the networks, ranging from 1 to 3. 2(b) shows the *camera view specific approach*, where the CNNs are initialized with the weights $W_{V_1}$ and $W_{V_2}$ obtained by pretraining with the corresponding view. $CNN_{V_1}$ would therefore only be initialized with weights $W_{V_1}$ that were obtained by pretraining with images only from $V_1$. 2(c) shows the *unified camera view approach*, where both CNNs are initialized with the same weights $W_{V_1 \cup V_2}$ obtained by pretraining with images from both views $V_1 \cup V_2$.

Table 1: Results of the experiments conducted solely on the synthetic dataset and evaluated by the metrics described in Section 5.3 regarding the impact of augmentation of the synthetic depth images. Note that training with augmented data can improve the results for all metrics.

| Metric | without augmentation | with augmentation |
|---|---|---|
| MAE | $0.23_{\pm 0.02}$ | $\mathbf{0.21}_{\pm 0.02}$ |
| Correlation $r_s$ | $0.85_{\pm 0.03}$ | $\mathbf{0.87}_{\pm 0.02}$ |
| Accuracy [%] | $85.29_{\pm 2.34}$ | $\mathbf{87.6}_{\pm 2.6}$ |
| Diag. Acc. [%] | $85.45_{\pm 2.77}$ | $\mathbf{86.96}_{\pm 2.47}$ |
| Sens. [%] | $91.5_{\pm 1.5}$ | $\mathbf{92.82}_{\pm 1.8}$ |
| Spec. [%] | $76.9_{\pm 6.01}$ | $\mathbf{79.0}_{\pm 5.88}$ |

### 5.2. Experiment 2

To answer the second question, we evaluated whether pretraining with the synthetic data improves performance when finetuning on real data (see Section 4.2). In addition, we compare these results with those of a model that was trained from scratch on the anatomical preparations dataset without any pretraining and with those of a model that was pretrained on ImageNet (Deng et al., 2009) and then finetuned with the anatomical preparations dataset. Furthermore, as in Section 5.1, we evaluated whether the augmentation of the depth data during pretraining has an influence on the results.

**Pretraining** There are four alternative ways of pretraining to initialize weights before finetuning. For training *from scratch*, without any pretraining, the models are randomly initialized. For the *pretraining on ImageNet*, both models $CNN_{V_1}$ and $CNN_{V_2}$ are initialized with the ImageNet weights. When pretraining using the synthetic dataset there are two possible ways, as this dataset also contains depth images of two views: In the *unified camera view approach*, both models are initialized with identical weights $W_{V_1 \cup V_2}$ obtained by pretraining with images from both camera views $V_1$ and $V_2$ of the synthetic dataset; see Figure 2(c). With the *camera view specific approach*, each model $CNN_{V_1}$ and $CNN_{V_2}$ is initialized with weights obtained by pretraining on synthetic depth images from only camera view $V_1$ or $V_2$ respectively; see Figure 2(b). This approach effectively halves the amount of pretraining data, but the finetuning is more specific. The pretraining on the synthetic dataset was performed as described in Section 5.1, but for the fact that the whole dataset was used as training set.

**Finetuning** The training from scratch and finetuning on the anatomical preparations dataset was performed with the same hyperparameters as in Section 5.1 in order to be comparable, using the separate network architecture; see Figure 2(a). As the anatomical preparations dataset consists of only two preparations, we trained on one preparation and tested on the other, and vice versa. The results are shown in Table 2.

Table 2: Results of the experiments without pretraining, with pretraining on ImageNet and pretraining with the synthetic dataset and subsequent finetuning on the anatomical preparations dataset, using the metrics and methods described in Section 5. Note that pretraining with synthetic data outperforms models without pretraining and with pretraining on ImageNet.

| Metric | from scratch | pretrained on ImageNet | pretrained on synthetic dataset | | | |
|---|---|---|---|---|---|---|
| | | | unified camera view | | camera view specific | |
| | | | wo/ aug. | w/ aug. | wo/ aug. | w/ aug. |
| MAE | 0.28±0.04 | 0.31±0.05 | **0.22**±0.04 | 0.24±0.04 | 0.23±0.03 | **0.22**±0.06 |
| Correlation $r_s$ | 0.88±0.04 | 0.9±0.03 | **0.92**±0.03 | 0.9±0.04 | 0.9±0.03 | 0.91±0.04 |
| Accuracy [%] | 79.25±6.98 | 77.37±8.22 | 89.03±5.51 | 86.91±6.14 | **90.45**±4.73 | 90.07±7.64 |
| Diag. Acc. [%] | 89.08±3.75 | 84.2±2.43 | 90.93±3.0 | 91.91±4.64 | **92.8**±2.14 | 92.43±4.05 |
| Sens. [%] | 91.21±4.23 | **93.79**±3.19 | 92.65±7.45 | 90.9±±0.29 | 87.64±4.51 | 91.16±6.72 |
| Spec. [%] | 88.66±4.61 | 80.05±4.6 | 89.84±5.49 | 92.24±6.59 | **95.34**±3.91 | 93.1±6.27 |

### 5.3. Metrics

When using two depth images $i_j^{V1}$ and $i_j^{V2}$ of the same pose from the two camera views as input, the continuous outputs $y_j^{V1}$ and $y_j^{V2}$ of the two models are averaged and compared with the corresponding quality label $x_j$. The following metrics are calculated: **Mean Absolute Error** (MAE), **Spearman correlation coefficient** ($r_s$) (see Equation (2) and Equation (3) in Appendix A), as well as two accuracies. The **Accuracy** measures how often the prediction differs with less than 0.5 from the label:

$$Accuracy = \frac{1}{N} \sum_{j=1}^{N} \mathbb{1}(|y_j - x_j| < 0.5)$$

where $y_j = (y_j^{V1} + y_j^{V2})/2$ is the average prediction, $N$ is the number of samples, $x_j$ is the label, and $\mathbb{1}$ is the indicator function. The **Diagnostic Accuracy** measures how often the prediction of whether a depth image is diagnostic or non-diagnostic is correct, i.e., whether label and prediction are both below or above the 2.5 threshold:

$$Diagnostic\ Accuracy = \frac{1}{N} \sum_{j=1}^{N} \mathbb{1}\left((y_j < 2.5 \wedge x_j < 2.5) \vee (y_j \geq 2.5 \wedge x_j \geq 2.5)\right)$$

Since it is worse to classify an image that is not diagnostic as diagnostic than vice versa, the **Sensitivity** and **Specificity** are also calculated for the diagnostic accuracy.

### 5.4. Results

The high accuracy of 87.6% in the experiment 1 based on the synthetic dataset (see Table 1) shows that synthetic depth images can be used to learn to assess poses and that training

with augmented synthetic depth images results in an improvement in all metrics compared to training without augmentation. Compared to the diagnostic accuracy of just 59.3% in case of a simple baseline that only predicts the mean of all labels, the proposed approach leads to a significant improvement. The results in Table 2 show that pretraining with the novel synthetic data improves training on real data. Despite the fact that the sensitivity of the diagnostic quality is highest (93.79%) for the model pretrained on ImageNet, the other metrics show, that there is no benefit from pretraining with ImageNet, compared to training from scratch. This suggests that the features learned on ImageNet are not useful for the task. However, pretraining on the novel synthetic depth images is useful since, except for sensitivity, all metrics are improved over those obtained without pretraining. Most importantly, the accuracy increases by 11 pp to 90.45% for the *camera view specific approach* without augmentation. With the same model, the diagnostic accuracy can also be improved by 3 pp.

Moreover, the *camera view specific approach* is performing slightly better compared to the *unified camera view approach* according to the more important accuracy metrics, which is presumably due to the higher similarity of the pretraining and finetuning dataset. In contrast to the results from Table 1, the augmentation of the synthetic depth images does not yield any benefits in these experiments. This is possibly due to an insufficient variance in the shape of the rather few anatomical preparations.

Note that we here obtain an overall accuracy of 90.45% when predicting diagnostic quality based on depth images. When comparing with the accuracy (93.0%) obtained for predicting the same diagnostic quality by using real radiographs (Mairhöfer et al., 2021), we can conclude that a similar quality assessment is possible with only depth images, although it is significantly more difficult.

## 6. Conclusion and Outlook

In this paper, we presented a framework for generating both synthetic depth images and radiographs from CT scans. We have shown that by pretraining on such a synthetic dataset relevant features can be learned, which are useful for the assessment of patients' poses on real data. This makes it easier to assess whether the patient's pose would lead to a radiograph with inadequate diagnostic quality before it is taken and thus protect the patient from unnecessary radiation due to a retake.

The main advantage of this framework is that the data acquisition problem, which is often critical in the medical context, can be solved by using already available CT scans. It is also possible to adapt the synthetic training data to different X-ray rooms and different ToF-camera setups in order to generate realistic and case-specific training data. Furthermore, it is possible to investigate which camera positions and how many cameras are best suited for the pose assessment task.

Although we have here only shown this for upper ankle joints, we believe that the framework can also be applied to other anatomies. Ideally, our novel way of generating radiographs and depth images can be used beyond the here demonstrated pose-assessment application.

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

## Appendix A. Metrics Definition

The **Mean Squared Error** (MSE) and **Mean Absolute Error** (MAE) are calculated as

$$\text{MSE} = \frac{1}{N} \sum_{j=1}^{N} (y_j - x_j)^2 \tag{1}$$

and

$$\text{MAE} = \frac{1}{N} \sum_{j=1}^{N} |y_j - x_j| \tag{2}$$

where $N$ is the total number of samples, $y_j$ is the prediction, and $x_j$ is the true label. The **Spearman correlation coefficient** $(r_s)$ is calculated as

$$r_s = 1 - \frac{6 \sum_{j=1}^{N} d_j^2}{N(N^2 - 1)} \tag{3}$$

where $N$ is the total number of samples, $d_j = R[x_j] - R[y_i]$, and $R(x_j)$ is the rank of $x_j$ and $R(y_j)$ is the rank of $y_j$.

## Appendix B. Framework Illustrations

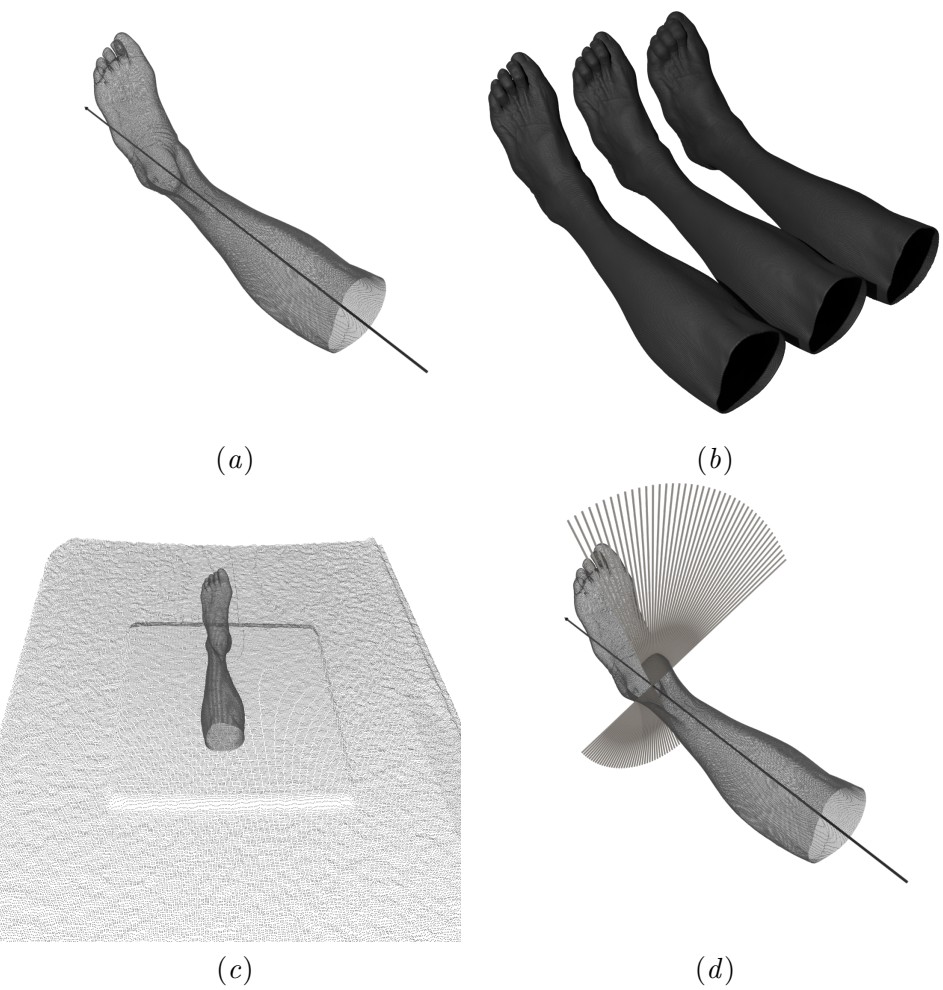

$(a)$ $(b)$

$(c)$ $(d)$

Figure 3: This figure illustrates the different steps described in Section 3. Figure 3($a$) shows the target anatomy cut out of the CT scan as a point cloud and the rotation axis, which passes through the center of the upper ankle joint. Figure 3($b$) shows the augmentation of the point clouds by moving the points along the direction of the normal vectors. Figure 3($c$) shows the target anatomy combined with the previously acquired X-ray room including the imaging table and detector. Figure 3($d$) sketches the positions of the X-ray device, which change due to the medial rotation around the longitudinal axis of rotation.

## Appendix C. Synthetic Example Images

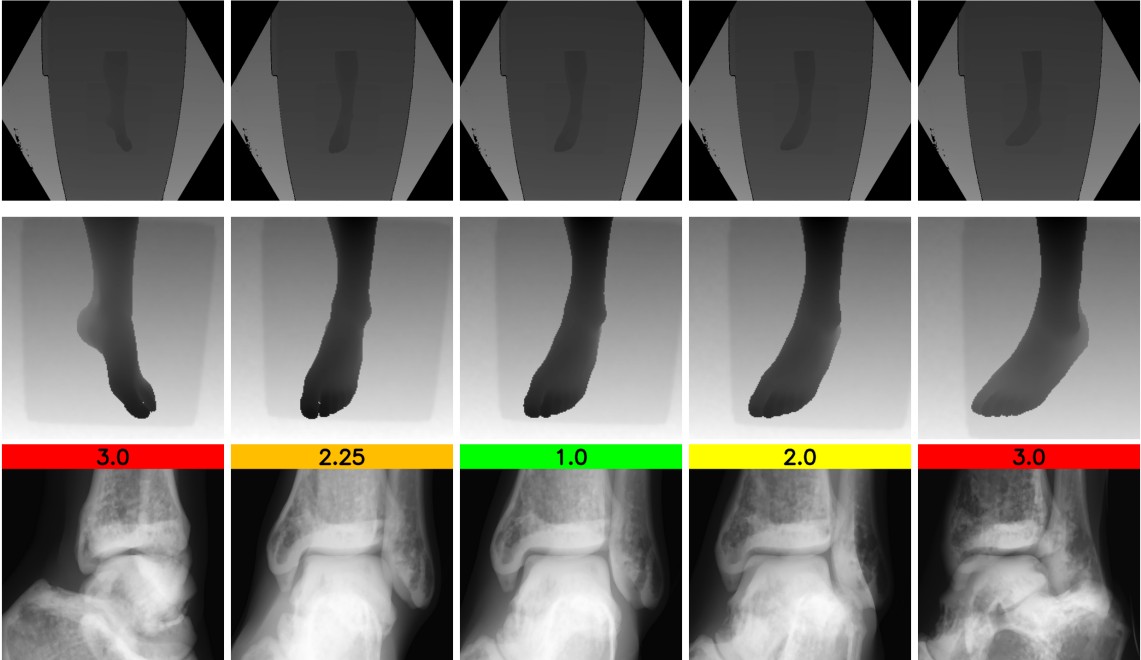

Figure 4: This figure shows synthetic images generated by using the framework. The first row shows the synthetic depth images that were generated with different rotations of the target anatomy. The second row shows the manually created ROIs for the synthetic depth images from the first row, which are then used for training. The third row shows the synthetic radiographs corresponding to the synthetic depth images, including their diagnostic quality, which has been assessed by the radiologists. Note that only small rotations are necessary to change a diagnostic quality of 1 to a diagnostic quality of 2, which is reflected in the visibility of the joint space in the different images.

