# OpenReview forum: "Synthetic Data Generated from CT Scans for Patient Pose Assessment"
_MIDL.io/2025/Conference — MIDL 2025 Poster_

### Official Review · Reviewer_EXhQ · 2025-02-21

**Confidence:** 4
**Preliminary Rating:** 3
**Final Rating:** 4

**Summary:**

The authors tackle the problem of estimating radiograph shot quality, given the pose of a patient, focused on the legs.
If proper leg placement can be optimized before making a shot, the quality of shots can be improved.

Therefore they create a DRR generation pipeline to create a radiograph dataset from CT scans to estimate poses of patients, focused on leg positions.
The method generates radiographs together with corresponding depth images.
There radiographs are then labeled by radiologists in order to create a dataset.

**Strengths:**

The targeted tasks seems quite simple but effective, determining the right position before making a shot makes a lot of sense. I imagine radiologists have some experience in this and can make good shots with experience, but I can imagine scenarios where standardising the approach with a method like this can really improve general quality of shots.

The main contribution is the pipeline of data generation to address tackling this task with only a few CT scans.

**Weaknesses:**

The method section is quite confusing, and after describing the dataset it is unclear how the actual experiments are set up and what is actually trained.
It seems that the network is trained to estimate quality given the depth images (which makes sense), but from there the actual method seems unclear.
The dataset part describes a scale of 1-3 (is this continuous?), and I imagine the model predicts this value.
How is the training setup done from there, what loss, optimizer etc is used.

It is hard to estimate really how well this pipeline approach works, since in the end it is a self-generated dataset and it is hard to tell whether an accuracy of 90% on the 1-3 assessment is 'good'.

Also the 'Anatomical Preparation Dataset' which is seems to be used for the final finetuning seems very small (2 legs from 2 patients?). This could be because it is somehow very hard to get?

**Detailed Comments:**

Please clarify the training setup more.

**Justification Of The Final Rating:**

The authors improved the paper and clarified the experiments further.  With that in mind, the weaknesses are alleviated a bit, it is still an experiment that can be hard to assess in how general this applies to other approaches, since it is a small dataset.

**Justification Of The Preliminary Rating:**

Pragmatic approach to a problem that could have some real benefit when solved.
However, the methodological design is lacking, and could be more clear and convincing.
There is no clear baseline to really compare this pipeline to, although this can also be because this problem could be unexplored in this specific setting.

**Questions To Address In The Rebuttal:**

Address the unclarities in the method, especially exactly how the training is set up, and how the output of the model is matched with the assessment score.

---

> ### Author Response · Authors · 2025-03-07
> **Response to Reviewer EXhQ**
>
> Thank you for your feedback, the comments and questions. We would like to address your concerns as follows:
> >The method section is quite confusing, and after describing the dataset it is unclear how the actual experiments are set up and what is actually trained.
>
> We have extensively rewritten and expanded Chapter 5 on Experiments and Results to better describe our methodological approach.
>
> >It seems that the network is trained to estimate quality given the depth images (which makes sense) [..] The dataset part describes a scale of 1-3 (is this continuous?), and I imagine the model predicts this value.
>
> Your assumptions are correct. We hope that the explanations in the updated manuscript are now clearer.
>
> >How is the training setup done from there, what loss, optimizer etc is used.
>
> In particular, we have also included the hyperparameters used for training.
>
> >It is hard to estimate really how well this pipeline approach works, since in the end it is a self-generated dataset and it is hard to tell whether an accuracy of 90% on the 1-3 assessment is 'good'.
>
> In experiment 2, (see Section 5.2) we show that the synthetic data generated with our framework can help to improve quality prediction on *real* data. If the synthetic radiographs differed too much from real ones, radiologists would not be able to label them regarding their diagnostic quality, and if the synthetic depth images were not comparable to real depth images, they would not be suitable for pretraining on the real images of the anatomical specimen dataset.
>
> In Section 5.4., we furthermore included the diagnostic accuracy for a simple baseline, which only predicts the mean of all labels, to show that our approach can indeed significantly outperform this baseline. We have also added information on the accuracy of the quality prediction on real upper ankle *radiographs* to show that our quality prediction based on depth images is not far off.
>
> >Also the 'Anatomical Preparation Dataset' which is seems to be used for the final finetuning seems very small (2 legs from 2 patients?). This could be because it is somehow very hard to get?
>
> You are absolutely right. Although a larger dataset of real data would be desirable, capturing anatomical preparations is not a scalable approach. This is also the reason why there are no other public datasets available for this problem. Therefore, we believe that the publication of our comparatively large dataset is of great value for further research in this area.
>
> >There is no clear baseline to really compare this pipeline to, although this can also be because this problem could be unexplored in this specific setting.
>
> Indeed, the lack of the baselines is due to the novelty in the approach. However, we have now included the two baselines mentioned above for comparison.

---

> ### Comment · Area_Chair_sVP7 · 2025-03-14
>
> Please update the final rating after appreciating the authors' responses.

---

### Official Review · Reviewer_ASxM · 2025-02-22

**Confidence:** 4
**Preliminary Rating:** 2
**Recommendation:** Poster
**Final Rating:** 4

**Summary:**

This paper describes a new method of using CT scans to generate synthetic X-ray imaging views and depth maps. The idea is to fisrt convert the CT scan to point clouds with optimizations to guarantee quality, followed by 3D pose augmentation to simulate imperfect patient orientations, and finally project the 3D model back to 2D via a virtual machine.

**Strengths:**

The main idea seems novel, by synthesizing X-ray images using CT scans, the lack of sufficient training data can be leveraged.

Experiment results seem convincing.

The overall length is adequate.

Figures are of good quality.

**Weaknesses:**

Related work section is unsatisfactory.

Lacking framework overview makes section 3 unsmooth.

Section 5.1 is hard to read and very confusing.

Section 5.2 needs improvement.

Language and formatting issues.

**Detailed Comments:**

Related work section needs improvement. A brief introduction of how the radiographs are reconstructed from CT scans is preferred. Methods in the cited references should be explained in details. Some abbreviations have not been explained, such as NURBS.

I would expcet an overall description of your proposal before diving into details.

HU means Hounsfield Unit, not Hounsfield (HU).

Section 5.1 is hard to read, I don't understand the setups in Fig. 2, what do you mean "pretraining on one view or both views"? Didn't you use random initialization or pretrained ImageNet, as mentioned previously? You said the model you used contains two neural networks per view, does it mean these networks were pretrained using ImageNet or any of the methods in Fig.2?

What's the representation of the model's prediction? You should present the metrices in Section 5.2 as equations and explain every symbol. Just like Fig. 2, you didn't even explain what ym and yn are.

**Justification Of The Final Rating:**

Most of my concerns in the review have been addressed except for some minor, non-technical issues like the usage of punctuation. I think the revised version meets the MIDL standard and thus recommend accepting.

**Justification Of The Preliminary Rating:**

The main idea of this paper is somewhat interesting. By synthesizing realistic data to pretrain quality assessment models the lack of training data can be leveraged. However there are several major weakness, including paper writing, experiment design, language issues. I would rate for weak reject for now unless the above issues are solved.

**Questions To Address In The Rebuttal:**

Related work

Section 5

Language issues

**Special Issue:**

No

---

> ### Author Response · Authors · 2025-03-07
> **Response to Reviewer ASxM**
>
> Thank you for the detailed feedback and the many helpful remarks! We would like to address them as follows:
> >Related work section is unsatisfactory: Related work section needs improvement. A brief introduction of how the radiographs are reconstructed from CT scans is preferred. Methods in the cited references should be explained in details.
>
> Thank you for pointing this out. We have substantially updated our "Related Work" chapter to better highlight and explain existing approaches for generating synthetic radiographs and extracting point clouds from CTs; see Section 2.
> >Lacking framework overview makes section 3 unsmooth: I would expcet an overall description of your proposal before diving into details.
>
> We have added a brief overview of the framework in Section 3 at the beginning.
> >Section 5.1 is hard to read and very confusing: I don't understand the setups in Fig. 2, what do you mean "pretraining on one view or both views"? Didn't you use random initialization or pretrained ImageNet, as mentioned previously? You said the model you used contains two neural networks per view, does it mean these networks were pretrained using ImageNet or any of the methods in Fig.2?
>
> Thank you for the remarks. In general, we use 4 alternative ways of pretraining: no pretraining, pretraining on ImageNet, the camera view specific approach, and the unified camera view approach. We hope that by restructuring the “Experiments and Results” Chapter 5 and making substantial additions to the text and figures, we have better explained our method and clarified your questions. If anything should be still unresolved after this update, please let us know.
>
> >Section 5.2 needs improvement: What's the representation of the model's prediction? You should present the metrices in Section 5.2 as equations and explain every symbol.
>
> We have expanded the metrics section and added more information about the calculation of the output of the models. We have also included formulas for the metrics we have developed. The formulas of the frequently used metrics can be found in Equations (1)-(3) in Appendix A.

---

> > ### Comment · Reviewer_ASxM · 2025-03-13
> >
> > Thank the authors for their efforts in the revised manuscript. All my concerns have been eased. Although there are still some minor issues like punctuation, I would recommend accepting.

---

### Official Review · Reviewer_Lobj · 2025-02-24

**Confidence:** 3
**Preliminary Rating:** 4
**Recommendation:** Poster

**Summary:**

This work deals with the generation of synthetic data from CT scans to improve patient pose assessment

**Strengths:**

It’s a well written paper that is easy to follow. While the core idea of generating depth and radiographic projections from CT data appears quite straight-forward; it does appear to be a novel application area for pose assessment– a quick search reveals literature in this space is virtually non-existent.

**Weaknesses:**

********************************************While not quite weaknesses, I hope the authors can respond to some of my comments in the detailed comments section******************************************

**Detailed Comments:**

You mention using 20 billion x-ray events in the simulation. Please explain the rationale for this choice and the cost of using more or fewer events and its impact on your experiments
It appears the synthetic data is created from 10 CT datasets. Considering the ready availability of CT scans / the relative ease of collecting them in appropriate settings, how would you expect your performance metrics to shift with the addition of even more synthetic training data.

**Justification Of The Preliminary Rating:**

The methods used are know to the community and dont involve any significant novelty. However, its a rather novel clinical use case with some potential to reduce recall rates. Worth highlighting to the broader community as as poster

**Questions To Address In The Rebuttal:**

Please consider addressing my comments in your discussion section

---

> ### Author Response · Authors · 2025-03-07
> **Response to Reviewer Lobj**
>
> Thank you for your feedback and remarks, we will address your questions as follows:
> >You mention using 20 billion x-ray events in the simulation. Please explain the rationale for this choice and the cost of using more or fewer events and its impact on your experiments.
>
> While more simulated X-ray beam paths would reduce the noise in the generated radiographs, the simulation time would increase. Fewer paths would result in noisier and potentially unusable radiographs. The use of 2·10^10 simulated paths allowed us to create realistic radiographs in a reasonable amount of time. The appearance of the synthetic radiographs was thoroughly assessed through close collaboration with the radiologists.
> We have included these explanations in Section 3.4.
>
> >Considering the ready availability of CT scans / the relative ease of collecting them in appropriate settings, how would you expect your performance metrics to shift with the addition of even more synthetic training data.
>
> Our results are already quite good using 10 CTs, which is probably due to the fact that we can extract a substantial number of depth images from only one CT (different poses, camera views, and augmentations). Furthermore, more training data could introduce a wider range of conditions, body types and anomalies, including various diseases and fractures, potentially enhancing the robustness and generalization capability of the model.

---

> ### Comment · Area_Chair_sVP7 · 2025-03-14
>
> Please update the final rating after appreciating the authors' responses.

---

### Author Rebuttal · Authors · 2025-03-07

**Rebuttal:**

We thank the reviewers for their valuable comments, questions, and appreciate the positive remarks. We have uploaded an updated version of the paper. Changes in the paper that relate to questions or comments from the reviewers are highlighted in red. In particular, the chapters "Related Work" and "Experiments and Results" have been completely revised and expanded. Due to the extensive nature of the revisions in these chapters, we have not marked every individual change to maintain readability. Instead, only the chapter titles have been marked.
We will respond to the individual reviews in order to address all concerns.

You can find the updated paper under Supporting Material.

**Supporting Material:**

/attachment/f3928ede5fb2de5a9460fe12e7846b10ce54c77c.pdf

---

### Author Response · Authors · 2025-03-14
**Response to Reviewers**

Dear reviewers,

we are grateful for the feedback we have received so far and hope that we could address all issues raised and answer all of your questions. Should there be any remaining points that require clarification, we would be glad to address them during the remaining discussion time.

---

### Meta-Review · Area_Chair_sVP7 · 2025-03-19

**Recommendation:** Accept (Poster)
**Confidence:** 4

**Metareview:**

In the initial submission, the major weaknesses were charakterized as the low amount of data used for the simulation (10 CT scans), unclear method and experiment descriptions that prevent proper assessment of the approach, and minor criticism on the overall clarity and structure of the manuscript.
The response provided by the authors addressed to a moderate degree of satisfaction the reviewers, to a point where all reviewers provide a weak accept rating. However, the major challenges on the overall implications of this work considering the small dataset prevail.

From my appreciation of the manuscript, the reviews, and the responses, I agree especially with Reviewer EXhQ's assessment that this approach is pragmatic and potentially useful, but that the limitations associated with the dataset and experimental setup are severe. However, as there is reviewer concensus for acceptance, I will concur.